# Smart Contract-Enabled Secure Sharing of Health Data for a Mobile Cloud-Based E-Health System

P. Chinnasamy [1], Ashwag Albakri [2], Mudassir Khan [3,*], A. Ambeth Raja [4], Ajmeera Kiran [1] and Jyothi Chinna Babu [5]

1   Department of Computer Science and Engineering, MLR Institute of Technology, Hyderabad 500043, India; chinnasamyponnusamy@gmail.com (P.C.); kiranphd.jntuh@gmail.com (A.K.)
2   Department of Computer Science, College of Computer Science & Information Technology, Jazan University, Jazan 45142, Saudi Arabia; aoalbakri@jazanu.edu.sa
3   Department of Computer Science, College of Science & Arts Tanumah, King Khalid University, P.O. Box 960, Abha 61421, Saudi Arabia
4   Head of the Department, PG Department of Computer Science, Thiruthangal Nadar College, Chennai 600051, India; arajacs1983@gmail.com
5   Department of ECE, Annamacharya Institute of Technology and Sciences, Rajampet 516126, India
*   Correspondence: mkmiyob@kku.edu.sa

**Abstract:** Healthcare comprises the largest revenue and data boom markets. Sharing knowledge about healthcare is crucial for research that can help healthcare providers and patients. Several cloud-based applications have been suggested for data sharing in healthcare. However, the trustworthiness of third-party cloud providers remains unclear. The third-party dependency problem was resolved using blockchain technology. The primary objective of this growth was to replace the distributed system with a centralized one. Therefore, security is a critical requirement for protecting health records. Efforts have been made to implement blockchain technology to improve the security of this sensitive material. However, existing methods depend primarily on information obtained from medical examinations. Furthermore, they are ineffective for sharing continuously produced data streams from sensors and other monitoring devices. We propose a trustworthy access control system that uses smart contracts to achieve greater security while sharing electronic health records among various patients and healthcare providers. Our concept offers an active resolution for secure data sharing in mobility computing while protecting personal health information from potential risks. In assessing existing data sharing models, the framework valuation and protection approach recognizes increases in the practicality of lightweight access control architecture, low network expectancy, and significant levels of security and data concealment.

**Keywords:** Health Data Sharing; IoT cloud; blockchain technology; smart contracts; proof-of work; confidentiality; integrity





## 1. Introduction

Currently, there is a resurgence of interest in using blockchain technology to build therapeutic and e-health facilities. With its distributed and reliable structure, blockchains have tremendous potential in several e-health sectors, such as the safe distribution of electronic health records (EHRs) and access control management among multiple therapeutic societies. Blockchains offer innovative ways to accelerate healthcare delivery, thus reinvigorating the healthcare sector [1–3].

With the proliferation of emerging technologies, especially mobile cloud computing and the Internet of Medical Things [4], the medical field has experienced substantial changes in e-health services. Patients can already access their personal information from home through smartphones and wearable devices, and then exchange information utilizing the cloud, where clinicians can access patient data and offer suitable health support. This

smart e-health service enables medical professionals to securely track patients, provide outpatient care at home, improve service delivery, and offer financial benefits to patients. Additionally, the availability of EHRs in clouds enables medical professionals to monitor patient health and provide adequate medical services through diagnostic and therapeutic mechanisms [4–6].

Fortunately, in addition to these benefits, the concept of data storage in EHRs has addressed various security concerns that impede the adoption of cloud-based e-health applications [6]. However, the secure sharing of health data between patients and healthcare providers remains a challenging task. Illegitimate agencies acquire intelligent methods to access EHRs without written permission, resulting in sensitive data and confidentiality breaches, as well as security attacks on e-health platforms [6]. Furthermore, patients find it difficult to monitor and manage the medical data shared in the cloud between different service providers. Consequently, it is essential to suggest effective access control strategies for platforms sharing cloud-based EHRs.

Conventional methods to access and control [7,8] the sharing of EHRs assume that requesters of cloud computing services are completely trustworthy, and allow cloud services to carry out all security controls and authorization responsibilities related to data consumption. Regrettably, this presumption no longer applies to portable clouds because they have sincere but inquisitive cloud servers. A crucial single point of failure for e-health connections can result from the fact that traditional authentication systems primarily depend on a predetermined entry point, that is, a centralized cloud server.

However, compared to traditional access control methods, blockchain-based access control offers several novel security mechanisms for e-health. A blockchain creates inalienable transaction ledgers as the first step in the intelligence sharing framework [9]. Consequently, transactions are only ever authored to the blockchain; rehabilitation behavior is not allowed. This indicates that activities documented in the blockchain cannot be changed or disrupted by any unit. This ensures a high level of reliability, stability, and credibility. Second, blockchain-based access control can obtain traceability and efficaciously address the problem of information leaks that can be brought on by intrepid servers. Third, authentication and user-verifying properties can be achieved by using proof-of-stake smart contracts [10]. Smart contracts can efficiently authorize access privileges to the stockpiling of health data by implementing strict rules regarding access. They can also efficiently identify and stop perceived risks to health connections in a decentralized manner. Finally, using central servers to guarantee fair treatment within and between complex interactions is eliminated when blockchain and smart contract innovations are combined. All networked agencies on the blockchain system will have a replica of smart contracts because they are public in the system, giving them equitable authority over all contract–operational processes.

We summarize our contributions in this study as follows:

1.  We propose a scalable EHR-sharing mechanism utilizing blockchain technology and the distributed storage of interplanetary file systems (IPFSs) [11] for a cloud-based Internet of Things (IoT) in e-health applications. We have developed a trustworthy access control system to enhance the protection of mutual healthcare data through smart contracts.
2.  The performance of the proposed method is analyzed through a compatibility test using a cloud-based IoT configuration. The findings of the evaluation showed that the recommended tactic is applicable to dissimilar e-health situations.
3.  Security assessments and several analyses using different evaluation criteria are presented to illustrate the effectiveness of the proposed system compared with existing e-health methods.

The remainder of this study is organized as follows. Section 2 discusses the work related to existing approaches to authorization management used for cloud-based EHR distribution. Section 3 discusses the blockchain principles and key elements in detail. Section 4 demonstrates the proposed framework model for a health data exchange method

based on blockchains and smart contracts, as well as the working model of an autonomous authentication mechanism for EHR exchange in a blockchain network. The implementation details are presented in Section 5, the performance evaluation is analyzed in Section 6, and the security evaluation is presented in Section 7. A discussion is provided in Section 8. Finally, the conclusions are presented in Section 9.

## 2. Related Work

There are many traditional solutions to solve the issue of the secure sharing of healthcare data in a cloud environment, such as mandatory access control, role-based access control, and trust-based access control [8]. Public key infrastructure has been widely utilized to provide secure authentication for healthcare data. In addition, some researchers have achieved fine-grained access control using different attribute-based encryption (ABE) methods [8,12]. Several studies have investigated blockchain technology's applicability in providing secure e-health data sharing. In [9], the authors adopted blockchain technology in an electronic healthcare system, e-health, to provide security services which can be shared among various users. Although the system proposed in [9] provides detailed explanations of the proposed framework, it lacks practical implementation, which is necessary to evaluate the proposed framework in real-world scenarios. Thus, several critical aspects of EHR sharing, such as adaptability, scalability, and access control have been overlooked. Furthermore, a data-processing approach based on blockchain technology was presented in [10] to ensure secure EHR exchanges across healthcare users. An autonomous solution based on blockchain technology was presented [10] to address the storage concerns related to the massive size of healthcare data. In [13], the authors presented a novel method that addressed the challenge of exchanging health data through cloud vendors. Smart contracts enforce an access control policy designed to restrict data access across untrusted parties, detect unauthorized entry into EHR stores, and enable healthcare data authenticity and monitoring.

The authors in [14] measured the solution layout on Ethereum's actual test platform. The results were evaluated using smart contract costing analyses and vulnerability analyses to indicate an increase in data administration, secure data searching impartiality, and privacy concerns. Studies in [15,16] designed systems that combined the IPFS platform with blockchain technology to offer information sharing in IoT scenarios to overcome the challenge of maintaining large datasets on the blockchain. These designs make it easier for IoT devices to communicate and share information in unreliable contexts. In [17], the authors proposed a secure access control platform to facilitate the monitoring of patients' vital signs generated by IoT devices. The proposed scheme utilizes smart contracts on the hyperledger fabric framework to authenticate users and securely share data in the blockchain network. Similarly, the scheme proposed in [18] utilizes blockchain technology to control access to an IoT network and assigns a penalty to a misbehaving user. Both [17] and [18] do not provide a full picture of where the data generated from numerous IoT devices are stored because the block size in any blockchain technology is limited.

Murala et al. [19] implemented blockchain technology in a virtualized environment to safely transport EHR data between authorized organizations. Blockchain technology is significant in such situations because it offers available data that can be disseminated to all businesses in the network. This method is contaminated because all health exchange information is recorded in blockchain technology as a hash function.

Zhang et al. [20] used pairing-based cryptography to create tamper-proof accounts of EHRs and enable physicians to incorporate them into blockchain operations. This can render patients' EHRs indisputable and shield them from inappropriate access. Using blockchain-based decentralized applications, researchers will also create safe payment protocols that allow patients and hospitals to pay dependably for diagnosis and storage services. Our proposed approach is effective and ensures security with minimal computational complexity, as shown by the security assessment and performance analysis.

Arani et al. [12] introduced the light-edge IoT device identification protocol, which uses a three-layer architecture consisting of an IoT device layer, a trust center at the network edge, and cloud service vendors. The findings demonstrate the exceptionalism of the proposed protocol over alternative strategies in terms of attack opposition, information sharing expense, and time expense. A detailed survey on data sharing in EHR environments using blockchain technologies was carried out by Kondepogu [21]. The detailed comparison of the existing work is clearly mentioned in Table 1.

**Table 1.** The comparison of existing research works.

| Papers | Techniques/Algorithm Used | Blockchain Types | Performance Evaluation Metrics | Limitations |
|---|---|---|---|---|
| [9] | Smart Contract | Permissioned Blockchain | Memory Analysis | It is not applicable for large pictures |
| [10] | BFT Consensus Algorithms | Private Blockchain | Communication Cost, Computation Overhead | Communication Overhead is high |
| [13] | Smart Contract | Ethereum Blockchain | Blockchain Cost | It is not applicable for Larger Datasets |
| [14] | ABE | IPFS | Computation Cost Analysis | Lacking to provide confidentiality |
| [15] | Smart Contract | IPFS | Computation Cost Analysis | It is lacking to offer integrity |
| [16] | Smart Contract | IPFS | Computation Cost Analysis | It is applicable for smart devices |
| [17] | Smart Contract | IPFS | Computation Cost Analysis | Lacking Cloud Security |
| [18] | Smart Contract | IPFS | Gas Cost Analysis | Lacking in Smart devices |
| [22] | Smart Contract | Public Blockchain | Computation Cost Analysis | Lacking in Access Control |
| [23] | Smart Contract | Public Blockchain | Registration Contract, Access Control Contract Cost | Lacking to offer high security |
| [19] | Smart Contract, Pairing-Based Cryptography | Ethereum Blockchain | Computation Cost Analysis | Lacking in Access Control |
| [20] | Edge-Cloud Environment | Amazon AWS | Communication Cost, Time Cost | Fail to provide insider attacks |
| [12] | Micro Genetic Algorithms | Cloud Environment | Optimization Cost | Fail to offer Confidentiality and Optimizations problems |

Despite the aforementioned research work, our proposed study's goal is to analyze access control over data uploads and distribution, and propose a mobile cloud management technique based on smart contracts and blockchains. The effectiveness of our proposed blockchain-based data access was evaluated in various real-world scenarios, including IoTs [24], vehicular network virtualization, and networking. Although previous related work lacks a discussion of EHR storage and management, the work in [17,22,23] marks a significant distinction between our research and previous EHR-sharing systems.

## 3. Background

We adopted the Ethereum blockchain framework [19] in the proposed scheme to construct an e-health system. Ethereum is closely related to other blockchain-based systems, such as Bitcoin, as a novel hybrid public blockchain. The benefit of adopting Ethereum is related to its efficient and versatile functionality, which allows developers to create various blockchain-based solutions such as e-healthcare.

The Ethereum blockchain maintains data blocks and executes smart contracts. Every block can be verified and then added to a chain to create a chain of blocks through consensus protocols, such as proof of work (PoW). Each block consists of a hash (unique value), checked proof of legitimate timestamp operations, and a preceding block hash. The preceding block hash is used to avoid altering the blocks or adding a block between them.

### 3.1. Ethereum Profile

Ethereum includes two distinct accounts: publicly managed and contractual accounts. Each profile was identified by an address of 20 bytes and specified by a public-key cryptosystem. Every user must create an account to join and communicate in the Ethereum network and become a networked agent [19].

### 3.2. Smart Contract

A smart contract is similar to a conscience-operating computer algorithm, which can be operated instantly if certain criteria are true. A smart contract is a crucial part of an Ethereum network. This includes the data and functions of blockchain transactions. Users can use their Ethereum wallet to communicate directly with smart contracts using binary interfaces. This asset helps organizations incorporate additional features, including data transfer and access control.

### 3.3. Transaction

A transaction is a data packet for transferring an ether from one wallet to another. In our proposed method, the data field is used to store user IDs, and a smart contract is used to check cloud-based data requests from users. The transaction structure is illustrated in Figure 1.

| |
|:---:|
| **Nonce** |
| **Address of Recipient** |
| **Gas Price** |
| **Gas Limit** |
| **Amount or Ether** |
| **Sender's Signature** |
| **Data** |

**Figure 1.** Structure of a Transaction.

### 3.4. Design Objectives

Our proposed framework must satisfy the following design criteria to allow EHRs to exchange information efficiently and safely on IoT clouds:

1. The framework must have user identification and authorization features. The system ensures that IoT device access is specifically checked to ensure trust in the scheme. This allows only authorized entities to control EHRs while attempting to prevent possible threats inside the EHR system.
2. The framework should have quick data recovery ability based on smart agreements via an authorization template. The architecture of user access control must be robust to prevent excessive communication overheads. Building a highly trusted framework with low operational costs is difficult when creating shared healthcare systems.

3.  The framework can help achieve strong security, system truthfulness, and data confidentiality, and provide IoT users with adaptability in creating the proposed solution for the most healthcare applications.

We suggest a trusted access control mechanism using a data sharing feature, including a smart contract, to handle authorization permission for EHRs to accomplish certain organizational objectives. A peer-to-peer IPFS-based cloud computing network has also been proposed to ensure distributed data processing and access to facilitate the sharing of EHRs.

## 4. Proposed System

In this section, a conceptual framework is presented with a focus on the method of data uploading and information sharing.

### 4.1. System Architecture

In the proposed healthcare system, all data are collected from a local network channel and stored in a public cloud, in which medical records are shared among authorized patients and healthcare providers. Medical records include patient data, patient ID, and area ID, indicating where the patient resided. Figure 2 depicts the proposed blockchain-based healthcare system.

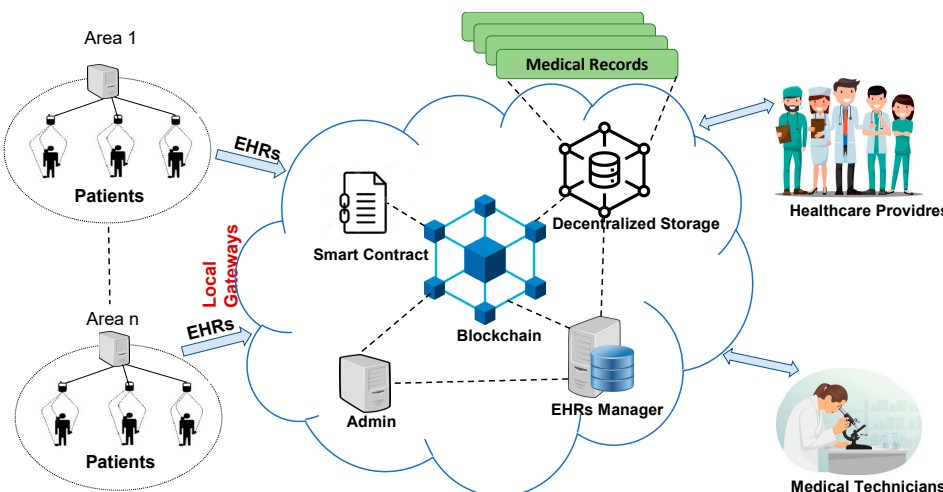

**Figure 2.** The Blockchain-based Healthcare System on a Mobile Cloud.

The proposed system is based on the following assumptions:

(a)  The data from the sensing devices are confidential and maintained by their regional users, that is, patients.

(b)  Electronic records were gathered from portable sensing devices through a web application incorporated into the mobile devices of the patients.

(c)  Healthcare provider actors have smartphones to access EHR in the cloud based on their hospital ID ($HP_{ID}$).

A patient's address on the blockchain is configured based on the area ID (AID) and the patient ID (PID) as Addr (AID, PID). As storing health records on blockchains is challenging because of the block size limitation, our goal is to keep patients' addresses on the blockchain, and a large number of medical records stored in the distributed file storage. In addition, a cloud provider, also called a health manager (HM), was introduced to handle medical files. Thus, if a user wants to access a patient's data on the cloud, he/she needs to remember the patient's address, which can be accessed mostly on the public blockchain. The design of the data flow model is illustrated in Figure 3.

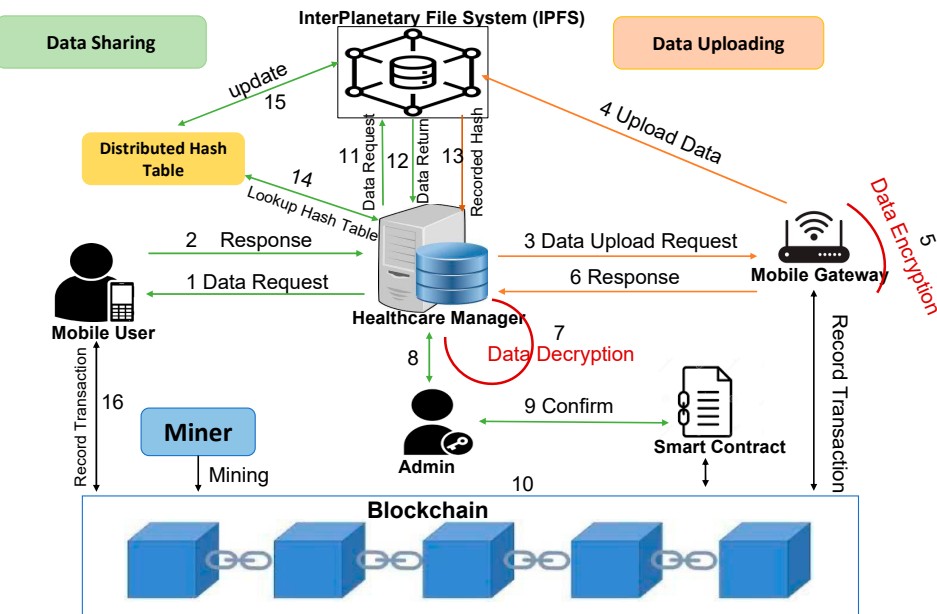

**Figure 3.** Design of the Data Flow Model.

The proposed IoT cloud-based blockchain network components are discussed below.

### 4.1.1. EHR Manager (HM)

The EHR manager is responsible for managing all transactions, uploading, and accessing data. HM administration features are controlled by smart contracts with strict user rules.

### 4.1.2. Admin

Admin is responsible for controlling all transactions and activities such as the attachment, modification, or revocation of access privileges. The admin is responsible for implementing smart contracts and is the only person capable of modifying or amending the rules in smart contracts.

### 4.1.3. Smart Contracts

The primary function of a smart contract is to monitor all legitimate activities within an access control framework. Messaging is carried out by the contract address and application binary interface (ABI) within the smart contract. In addition, they can categorize, authenticate/validate transactions, and grant access privileges to patients by activating transactions or messages. Blockchain actors, minors, and admin can access smart contracts and their activities.

### 4.1.4. Decentralized Storage

Blockchain technology is not designed to exchange and store large amounts of data; we used an interplanetary file system (IPFS) to overcome this limitation. This is an optimistic approach for creating a file storage framework in a blockchain system. The IPFS and Kademlia decentralized hashing tables (DHT) are constructed on top of the BitTorrent protocol. In the IPFS system, the centralized function is removed, and clients can process data on the same file server in a wide range of network master nodes with benefits over the traditional data storage system, including no single points of failure, high storage space, and enhanced data recovery.

In our proposed method, all health records are encrypted using the improved key generation scheme of the RSA (IKGSR) algorithm [20] and securely locked onto IPFS subsystems. The hash values of the health records are stored and managed in a DHT, mostly by the health manager. Furthermore, the assimilation of a smart contract with

the IPFS helps to enhance cloud storage, thus providing efficient data sharing and access management. The EHR storage system runs inside the IPFS, in which patient data are maintained by an individual network node.

### 4.1.5. Block Structure

The components of the block structure are shown in Figure 4. Transactions in a block are structured in a Merkle tree-based model in which the leaf vertices represent a smartphone-shared data operation. To make a prescription for the results, a smartphone user must provide a patient's address (Area ID, Patient ID) to create a transaction that is authorized after a certain timeframe with the user's personal key. The certificate authority attempts to build relationships between the users and cloud servers. The sequence flow of the proposed method is illustrated in Figure 5.

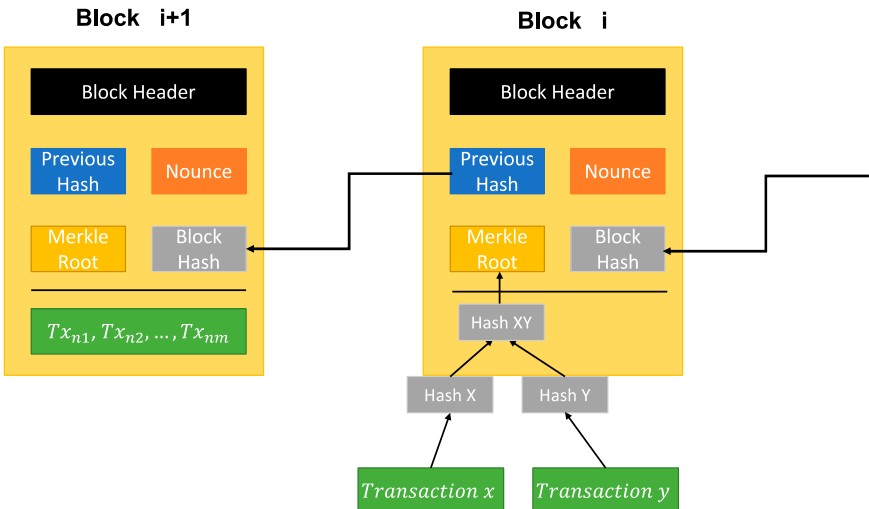

**Figure 4.** The Structure of a Block.

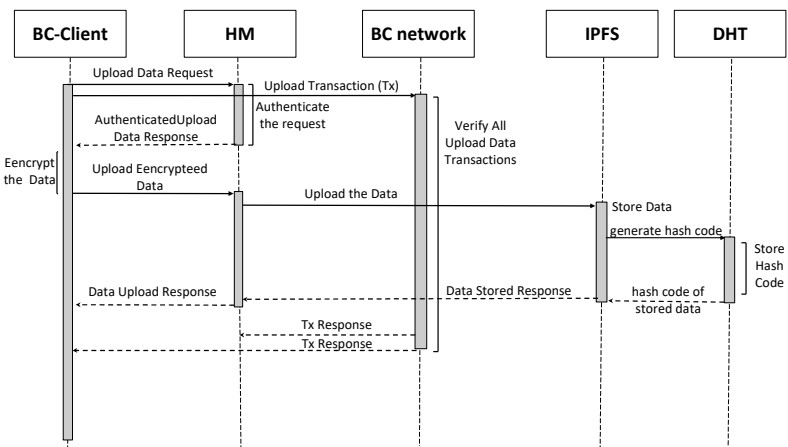

**Figure 5.** The Sequence Diagram of Activities of the Proposed Method.

### 4.2. Process of the Proposed Work

In this section, we implement secure data-upload and data sharing mechanisms by utilizing smart contracts and blockchains. The use of smart contracts enables high productivity and transparency in the automated management of blockchain transactions. We built decentralized user modules for smartphones; the decentralized modules provide basic functionalities to users to interact with the blockchain network throughout the cloud. This layer is responsible for encoding transactions and data demands, generating digital signatures for transactions, and connecting to the blockchain to monitor transaction activities. In

addition, the user interface module is intended to communicate with the user and request a controller unit to manage the user request information. The workflow of the operations for uploading and exchanging EHRs is shown in Figure 6. The description of every sequence of iterations is as follows:

1. A mobile adapter instantiates a query to migrate the data to a cloud server as a new block.
2. The blockchain client, the user, evaluates and provides the query for authentication to an administrator of the EHRs, as well as the consensus mechanism.
3. The administrator of the EHRs authenticates the application through a consensus mechanism with a tight security rule. If a request is approved, a notification of the uploaded data is forwarded to the firewall.
4. The firewall can now pick the EHR, a health data file gathered via smart technologies, and encrypt the data using the EHR owner's key pair. The firewall automatically releases encrypted messages to cloud services using the IPFS.
5. IPFS management stores the file system in a user device by employing uploaded details, and dynamically generates a hash code stored in the DHT database.
6. Copying accounts are organized into data items and incorporated into transaction repositories to validate the ledger by mining.
7. The uploaded operation is modified for monitoring by a wireless terminal.
8. The mobile user executes a query containing the ID and signature of the secret key of the server.
9. The blockchain application node provides an authenticated user grant of access to upload the data.
10. The EHR manager can check the user's legal entitlement through the hashing algorithm access management plan. If the query for permission is authenticated, the administrator of the EHRs should review the IDs and then forward the requests to the IPFS repositories for data retrieval.
11. EHR management requires an asymmetric encryption algorithm to decrypt the required information and return it to a recipient.
12. The recipient's smartphone is updated with the uploaded transactions for monitoring through the blockchain application.

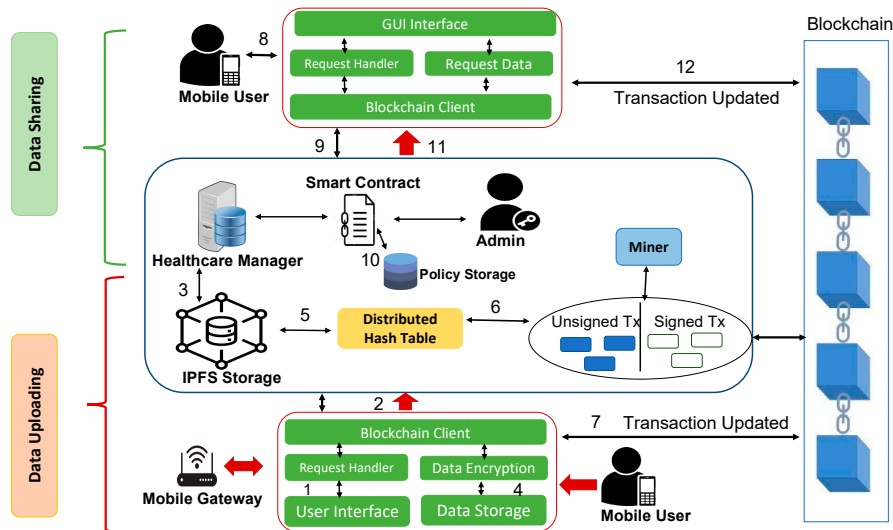

**Figure 6.** The Proposed Method Data Uploading and Sharing.

### 4.3. Uploading and Sharing Processes

In this section, the details related to uploading and sharing processes are explained.

### 4.3.1. Uploading Process

The procedure for exporting the EHRs is shown in subsequent steps:

Step 1:   Mobile Gateway execution

The mobile adapter creates a blockchain username and password to access the Ethereum blockchain, and then authorizes a query to upload a transaction. The web interface connects the user node blockchain to nodes in the cloud. These configurations establish a processing request (Ts) within the data-uploaded process as a query that contains application information.

Step 2:   Data Uploading to IPFS Storage (Performed by Healthcare Administrator)

The EHR supervisor issues commands to the smart contract, such as a reminder of a fresh data-uploading invitation, followed by acquiring a transaction again from gateways. The smart contract validates transactions based on the policies listed in the policy repository. When the transaction is authorized, a response is sent to the gateways through blockchain clients/users, allowing information to be uploaded (Step 3 in Figure 5).

When a transaction is authorized, the mobile gateways can use an asymmetric encryption technique to encode the EHR files using the public key of the EHR administrator (Step 7 in Figure 6). Subsequently, the encrypted information is posted to the IPFS cloud platform, which is linked to the gateway's blockchain web application (Step 4 in Figure 6).

Step 3:   Mining the Transaction

In addition to sending EHRs into cloud storage, the information from the publishing communications (Area ID, Patient ID) and hash value is stored in the unverified transaction pool. Miners group transactions throughout the pools into chunks for computing regularly. The identity of the miner is sent to other miners to confirm the block's data. When all the miners approve, the authenticated block and its signatures are chronologically added to the blockchain. Subsequently, all service providers acquire that transaction and utilize the blockchain application to replicate the blockchain copies (Steps 6, 7, and 8 in Figure 5).

### 4.3.2. Data Sharing Process

Mobile users can access information from the cloud through an information sharing mechanism that incorporates access control. The steps of the procedure are as follows.

Step 1:   Request Generation (Performed by the Mobile User)

A user must create an online transaction (Tx) as a request message to view data from the cloud. Members can view our blockchain by establishing a blockchain profile with a secret key for signing the transaction and a public key for the user's identity. To receive data, users must also provide the region's address via metadata for the transactions. Subsequently, the transaction (Tx) is forwarded to the cloud based on EHR management for authorization and verification (Steps 8, 9, and 10 in Figure 6).

Step 2:   Data Retrieval

The EHR manager receives the client transactions for auditing and validation. The EHR management immediately sends a message to the smart contract to validate the transaction. The smart contract verifies and sends a message to the EHR management based on the policies table, including user authentication tokens. However, if a user's public key is found in the policies table, the requester's access authorization is approved and the desired information can be accessed. Otherwise, the smart contract imposes penalties on the submitter and refuses the recipient's future requests (Step 10 in Figure 6).

Step 3:   Sharing Transactions to Blockchains (Performed by Miner)

Transactions are organized as bundles and sent to a transactional pool for processing by a group of miners, similar to how data are uploaded. All authenticated transactions are documented and added to the blockchain, available to all smartphone users. The blockchain application incorporated with the end user's device allows users to update the contents of their operations (Step 12 in Figure 6).

### 4.4. A Real-World Case Study Scenario

In this section we demonstrate an application of our proposed scheme in a real-world scenario. Figure 7 depicts a hospital information system scenario in which users such as patients, insurance companies, government agencies, and smart devices exchange data. IKGSR key generation, which produces the key pairs, initiates a sample set of circumstances workflow. Each smart gadget sends its attributes to the key generation authorities to register. After authentication, IKGSR sends the relevant device the Public Key. The device can use this key to generate the access code and encode its contents using the IKGSR encoding function using a blockchain-based smart contract. The medical supplier receives the uploaded encrypted files and access key. At that moment, the patients or smart devices send a query to HIS, and HIS responds by sending the patients or smart devices encrypted information and an authentication key. Patients or smart devices must finish the decoding procedure in order to access the encrypted message. The steps in deciphering are comparing the access key, trying to match the characteristics of the system administrator, and confirming the name of the system administrator. The adversaries or users won't be able to access the actual data of the owner when any of the three requirements are not met.

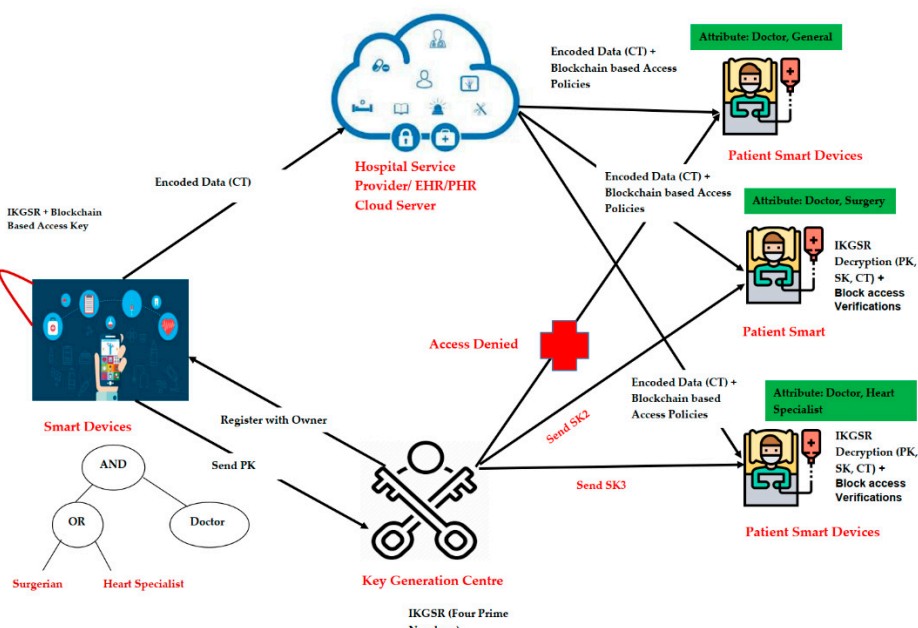

**Figure 7.** Application Scenario of the Proposed Method.

### 5. Implementations

To investigate and validate the proposed system, we present evidence for the framework development of federated access control for exchanging EHRs on mobile clouds. The subsequent sections disclose information on the integration of the configuration management.

### 5.1. System Testing Environment

We adopted a platform to exchange EHRs on a mobile cloud. The Ethereum network node is configured on Amazon Web Computing (AWS) [25], where two AWS computing node VMs [26,27] are used as miners, and two Ubuntu 16.04 LTS VMs have been acting as administrators and EHR managers. The contract was created and implemented using a Solidity scripting language [27]. Every blockchain communication is executed by web3.js. People can interrelate smart contracts via a mobile device in which a Geth client [28] is configured to turn each device in with an Ethereum node. The web3.js toolkit [29], a compact Java framework for integrating smart contracts and blockchains, was incorporated into developing a mobile application that connects to the Ethereum blockchain network [30].

Throughout this study, we investigated the consequences of EHR distribution using two Honor devices running on Android OS type 7.0.

### 5.2. Design of an Access Control Mechanism

We created a smart contract to establish the proposed access control mechanism. In addition, we created a central component that depicted the working process of the EHR distribution system, as shown in Algorithm 1.

---

**Algorithm 1.** Proposed Access Control Algorithm

---

**Input:** *Transactions (Tx)*
**Output:** *Authorized or block*
**Initialization:** *EHR Owner*
1: New transaction *Tx* from patient
2: Collect the Pu of the patient $PK \leftarrow Tx.getSenderPublicKey()$
3: Send the public key to Admin for verification
4: **if** *PK* is available in the policy list
    $pL(PK) \leftarrow true$
5: **end**
6: $decodedTx \leftarrow chinnaDecoder.decode\ (Tx)$
7: $Addr \leftarrow web3.eth.getData(decodedTx([DataIndex])$
8: Specify *AreaID* and *PatientID* $\leftarrow A_{ID}\ Addr(Index[A_{ID}]); P_{ID}\ Addr(Index[P_{ID}])$
**EHRs Retrieval** *(Consensus Algorithm)*
9: **while** true **do**
10:    **if** $pL(A_{ID}) \rightarrow true$ **then**
11:        **if** $pL(P_{ID}) \rightarrow true$ **then**
12:            $Result \leftarrow retrieveEHRs(PK, Addr)$
13:        break;
14:        **else**
15:        $Result \leftarrow penalty(PK, action)$
16:        break;
17: **else**
18:    $Result \leftarrow penalty(PK, action)$
19:    break;
20: **end**
21: **end**
22: **return** *Result*;

---

To observe the transaction activity of the smart contract, we first constructed a data sharing contract managed by administrators. In the blockchain, we refer to the recipient's public key as PK, the recipient's role as userRole, and the recipient's address as Addr.

The following tasks are performed by the contract:

- AddUser (PK, userRole): This was processed using Admin. This procedure enables new users to be added to the primary contract. The user is authenticated by his/her public key and has been assigned a position throughout the contracts depending on his/her requests. User information is recorded and stored as an element of the database server.
- DeleteUser (PK, userRole): This is executed by Admin, who can remove the user. It is used to eliminate user access based on the public key that corresponds to the user. Consequently, all private details were removed from the data storage.
- PolicyList (PK): A sequence of policies executed by admin. A colleague of a healthcare professional and patients may consent to a protocol that describes the professional relationship. For instance, a patient may have a single doctor or several physicians who need access to a patient's electronic health records (EHRs) to provide suitable healthcare services. When a smart contract executes transactions, policy reports consisting of all parties' public keys are verified.

- EHRsRetrieve (PK, Addr): This is executed by an EHR manager who enables health records to be retrieved from cloud services. The smart contract requires a networking operator to supply the physician's geolocation, Patient ID, and Area ID. The contracts subsequently evaluate the information and send communication to the EHR administrator, instructing him/her to retrieve and provide information to the applicant.
- Punishment (PK, action): This is executed by the administrator; whenever the EHR administrator detects an unlawful demand from the system, the contract is notified, and the user is penalized. Consequently, we issue an alert signal to an illegal mobile person in public.

## 6. Performance Evaluation

The performance of the proposed framework is evaluated based on two criteria: the average time required to process the user's requests on the cloud and the time required to access the users' data.

The average time required to evaluate several access user requests in the cloud was calculated (Figure 8). Compared to the approach without an authentication mechanism, our proposed system, which encompasses access control on smart contracts, takes longer to execute service requests. The time spent on identity management and authorization confirmation was added to this cost. However, the increased cost was approximately 5 s in the worst-case scenario—50 queries. Thus, it is considered to be minimal and appropriate for real-world situations. This result demonstrates the compact architecture of the proposed model for user authentication and authorization management.

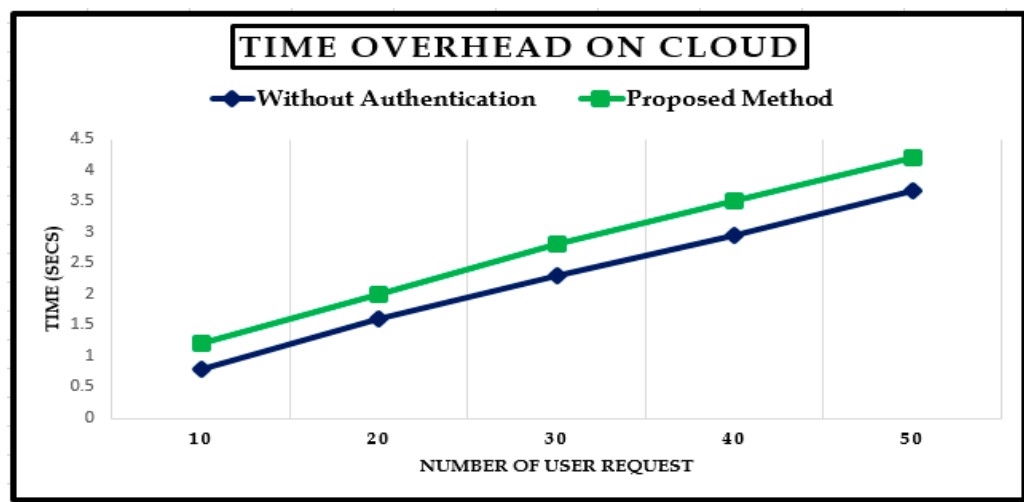

**Figure 8.** Average Time Overhead Based on the Number of User Requests.

In addition, we demonstrate the capability of our EHR distribution system through the computational complexity of acquiring EHRs for the IPFS blockchain cloud storage (Figure 9). We analyzed the amount of time it took to complete the EHR access process, from the time of demanding data access to the time of obtaining information on a smartphone. In addition, we investigated the time efficiency of EHRs using a blockchain-based repository, IPFS, versus a central server supported by a simple storage service (S3). The findings of the evaluation, considering various large EHR files, reveal that our distributed storage solution on the blockchain is faster (takes less time) than the centralized storage strategy to access data.

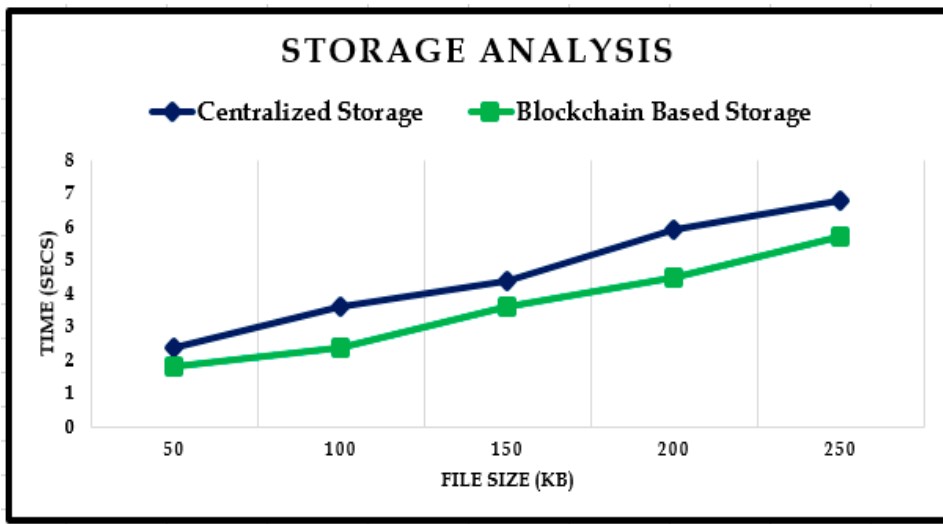

**Figure 9.** Time Required to Access the Data on the Cloud versus the Blockchain.

Compared with traditional cloud designs, the network latency measurement shows that our design achieves a compact EHR distribution with low computational latency. The second conceptual aim is explained in the performance section.

*Comparison of Encryption and Decryption Time*

We have already introduced the IKGSR [31] algorithm to enhance the security of asymmetric algorithms in cloud environments. In the proposed environment, we tested IKGSR against the traditional RSA algorithm. Figures 10 and 11 clearly show that IKGSR requires more time to encode and decode input health data. However, the IKGSR algorithm offers high security as well as security against chosen ciphertext and timing attacks, as shown in [31].

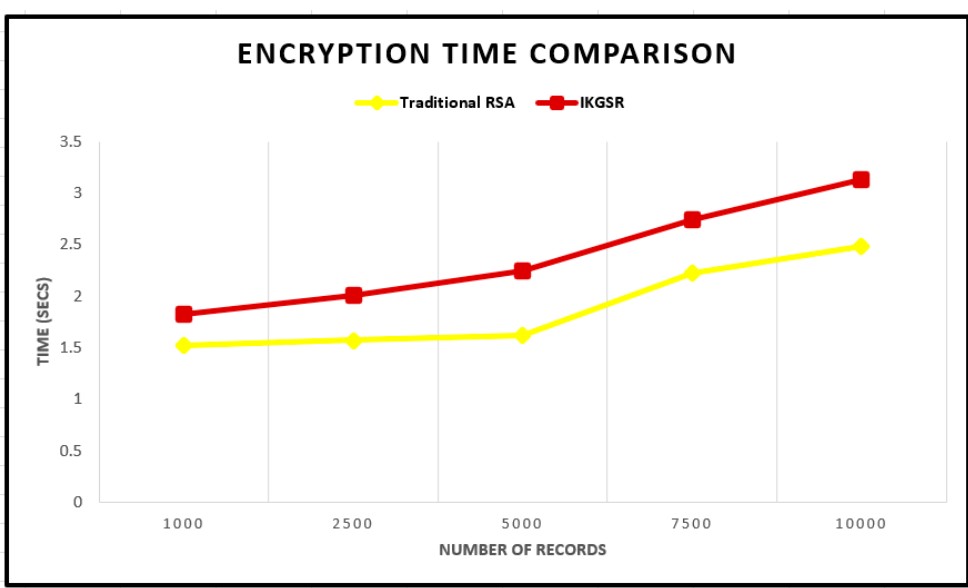

**Figure 10.** Encryption Time Comparison against Traditional RSA.

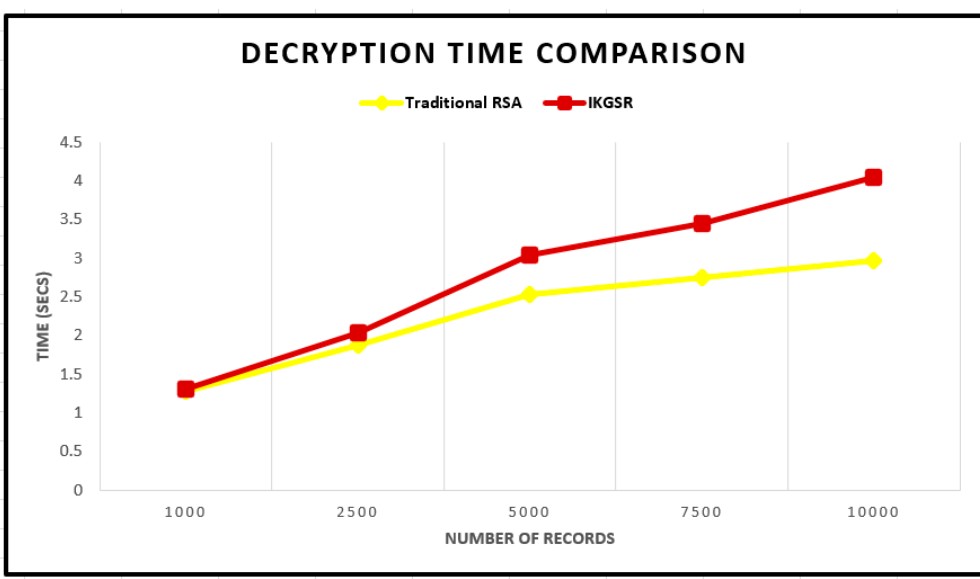

**Figure 11.** Decryption Time Comparison against Traditional RSA.

## 7. Security Evaluation

Security is a major concern when implementing healthcare applications using various technologies; where sensitive data are protected from unauthorized access, the integrity of the data is enhanced, and the privacy of the users is enhanced. These assertions were used to demonstrate the trustworthiness of the proposed approach.

In this section, a security analysis of the proposed framework is conducted using different attack situations. In addition, the advanced features of the suggested data exchange method are addressed to emphasize the utility and practicality of our approach.

**Theorem 1.** *If an adversary has access to information in the cloud platform without permission granted by the EHR administrator, retrieving and reading health information through our systems is exceptionally difficult.*

**Proof.** All health records were encoded with the public key of EHR management before being uploaded to decentralized cloud services based on our proposed architecture. To receive data from the cloud, recipients must have access to the e-health owner's secret key to decode original information. This secret key is unique, and only the EHR management can access it. Consequently, guessing the secret key to decode and acquire information in cloud storage is extremely difficult for an adversary. □

**Theorem 2.** *Assume that an attacker can request cloud e-healthcare data through suspicious transactions. It is difficult for an attacker to gain authorization by tampering with the access control architecture.*

**Proof.** To demand an approved transaction, a mobile node might need a private key (SK), public key (PK), and request ID, which are required to authenticate the transaction when it is sent to cloud services using the authorized (Tx) and SignSK syntax (rawTx PK, requestID, and timestamp). Only the user's public key is accessible to many individuals throughout the blockchain network, whereas the user's secret key is accessible only to the owner/user. Consequently, an attacker cannot authenticate transactions for database access using such a secret key. Additionally, because miners would remove any erroneous records from the Ethereum blockchain, the access management system would be impervious to any potential dangers. □

## 8. Discussion

The suggested EHR sharing system was reviewed and assessed using a variety of assessment measures to demonstrate that our concept is feasible in real-world circumstances.

### 8.1. Adaptability

Because our solution is built on a portable device, everyone with a smartphone can interact with it, providing users considerable flexibility and convenience. Our solution is compatible with most portable devices, such as iPhones and Androids, thus enhancing the accessibility of our solution in several different disciplines.

### 8.2. Availability

Authorized mobile users can download e-healthcare data at any time and location using a mobile application. Using a smartphone device enables users to access the system and the health data stored in the cloud.

### 8.3. Avoid Single Point of Failure

In our architecture, the distributed storage option of the IPFS is utilized, resulting in the resolution of a single point of failure issue. Additionally, even with a blockchain solution, access control is dispersed across multiple groups in a peer-to-peer manner, which can help solve this problem.

### 8.4. Integrity

Integrity ensures that medical data is exchanged across legitimate users in an unaltered state. To prevent tampering with the data, health records gathered through mobile platforms are often encoded, that is, IKGSR is utilized. Furthermore, mobile phone users remain unable to edit signed transactions with smart contracts, and no parties are allowed to manipulate the meaning of transaction records. Note that, in our model, smartphone users may not be able to amend or modify smart contractual agreements or access controls.

### 8.5. Data Security

Our access control approach ensures confidentiality and user data security by utilizing smart contracts' security capabilities. Illegal access is limited by user identification and smart contract permission, which prohibits significant threats from gaining access to cloud services. Furthermore, the consensus algorithm rejects and deletes illegitimate transactions by using our smart contract. Another intriguing aspect of our approach is that all blockchain parties have identical information management responsibilities and are responsible for monitoring all communication. Consequently, any changes to cloud patient files can be immediately identified by phone users and reported, ensuring the protection of medical information.

Our scheme can be protected against potential assaults and threats by conducting a thorough review of numerous design requirements. Our architecture also has some appealing advantages such as adaptability, high system reliability, and privacy protection, all of which are valuable in application scenarios.

Furthermore, we demonstrate an evaluation of our proposed design with other significant findings in Table 2 regarding various innovative elements by employing two choices: Y (yes–present) and NA (Not available). Depending on the detailed related work described in Section 2. The evaluation results show that the proposed approach outperforms traditional methods, making it a viable option for upgrading existing e-health solutions.

**Table 2.** Performance evaluation comparison with existing methods.

| Parameters | [15] | [16] | [18] | [22] | [12] | Proposed System |
|---|---|---|---|---|---|---|
| Flexibility | NA | NA | NA | NA | Y | Y |
| Availability | NA | NA | NA | Y | Y | Y |
| Access Control | NA | Y | NA | Y | Y | Y |
| Identity Access | Y | Y | Y | Y | NA | Y |
| Authentication | Y | Y | Y | Y | Y | Y |
| Data Privacy | Y | Y | Y | Y | Y | Y |
| Integrity | Y | Y | Y | Y | N | Y |

## 9. Conclusions

This study suggests a new EHR sharing strategy that utilizes cloud-based services and blockchains. We highlight the major shortcomings of existing EHR sharing platforms and provide cost-effective solutions through a real-world simulation experiment. This project aims to provide a reliable access control system that relies on a smart contract to regulate access rights and ensure efficient and reliable EHR information exchange. We implemented an Ethereum blockchain only on the AWS cloud to evaluate the effectiveness of the suggested solution; healthcare organizations can interface with both EHR-centralized repositories through the created mobile application. To accomplish secure information management and sharing, we combined the peer-to-peer review process to integrate seamless storage solutions using a blockchain. Compared with traditional approaches, the operational results show that our architecture could require doctors/users to transmit medical information in an adequate and accurate fashion through mobile cloud settings. Our access control system can efficiently detect and block unlawful access to e-health systems, ensuring patient confidentiality and computer security. We provided security analyses and assessments of the technological contributions of the proposed system, demonstrating the benefits of our approach over existing alternatives. Our distributed ledger system is a milestone towards the efficient management of e-health information in portable computing, which would be interesting across many smart healthcare systems.

**Author Contributions:** Conceptualization, P.C. and A.A.; methodology, M.K.; validation, A.A; resources, J.C.B.; data curation, A.A.R. and A.K.; writing—original draft preparation, P.C. and A.A.; writing—review and editing, M.K. and A.A.R.; visualization, J.C.B.; supervision, P.C., J.C.B. and M.K.; project administration A.A., J.C.B. and A.K. All authors have read and agreed to the published version of the manuscript.

**Funding:** This research work is funded by the Deanship of Scientific Research under grant number W44-83, Jazan University, Saudi Arabia.

**Institutional Review Board Statement:** Not applicable.

**Informed Consent Statement:** Not applicable.

**Data Availability Statement:** Not applicable.

**Acknowledgments:** The authors extend their appreciation to the Deanship of Scientific Research at Jazan University in Saudi Arabia for funding this research work through the grant number W44-83.

**Conflicts of Interest:** The authors declare no conflict of interest.

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
