# Peer review of "Smart Contract-Enabled Secure Sharing of Health Data for a Mobile Cloud-Based E-Health System"

_applsci, doi:10.3390/app13063970_

Round 1
Reviewer 1 Report
-The authors introduced a new EHR-sharing strategy that utilizes
cloud-based services and a blockchain to achieve greater security while sharing electronic health records (EHRs) among various patients and healthcare providers
-The following major corrections are required:
Section 1: Introduction
-The introduction section is too general, and it introduces concepts that are well known about the scalable EHR sharing mechanism utilizing blockchain technology. The introduction does not stimulate to go ahead with the remaining of the paper because it does not introduce any really new topic/solution. Furthermore, "the research motivation" at the introduction section is missing. Please rewrite tis section.
Section II: Related work
-In this section, the authors should be describe some of the research works about the scalable EHR sharing mechanism utilizing blockchain technology, while some of the papers that should have be included are:
https://link.springer.com/chapter/10.1007/978-3-031-22835-3_4
https://link.springer.com/article/10.1007/s12652-020-02561-3
https://www.sciencedirect.com/science/article/abs/pii/S1389128621004916
https://ieeexplore.ieee.org/abstract/document/9335011
https://ieeexplore.ieee.org/abstract/document/9788462
https://link.springer.com/article/10.1007/s10586-020-03152-9
…..
-In addition, a conclusion of related work in the forms of a table in terms of utilized technique, evaluation tools, data set, performance metrics, advantages, and disadvantages could reconcile from other researchers work to the own one.
Section IV: THE PROPOSED SYSTEM
-Please provide a sequence diagram to show the interaction between components of the proposed architecture according to Figures 3 and 5.
-What is the overhead (time complexity) proposed solution? Please provide a subsection to discuss about the overhead (time complexity) of proposed.
-Please provided real-world case study example for better understanding the proposed approach in more details.
-Your proposed solution is very similar to the following papers. What is the difference between proposed model and mentioned papers?
https://ieeexplore.ieee.org/abstract/document/8717579
Section VI: PERFORMANCE EVALUATION
-The evaluation is incomplete. I would like to see an evaluation on the proposed solution in terms of execution time under different scenarios.
-Paper needs some revision in English. The overall paper should be carefully revised with focus on the language: especially grammar and punctuation.
-Overall, there are still some major parts that the authors did not explain clearly. Some additional evaluations are expected to be in the manuscript as well. As a result, I am going to suggest Major revision the paper in its present form.
Author Response
Dear Reviewer,
First, we would like to thank you for your valuable comments. We confirm that we have taken all their comments into our consideration when updating the paper. Kindly find the attached file.
Thank You
Regards,
Dr. Mudassir Khan
Corresponding Author

Reviewer 2 Report
This work presents an integrated information system based on distributed ledger technology. The scientific task it resolves is complex, and results are related to implementation, efficiency, security, and trustworthiness. This is a consequence from applied diverse technologies.
The authors research achievements on the subject via a preliminary study of the literary sources.
In the proposed healthcare information system, all data are collected from a local network channel and stored in a public cloud where medical records are shared among authorized patients and healthcare providers.
The goal of the solution is to keep patients' addresses on the blockchain, and the vast number of medical records kept on the distributed file storage. The primary function of a smart contract is to monitor all legitimate activities within an access control framework.
The proposed architecture is presented convincingly through the roles, and data flows and processes of their processing.
Validation evidence for the proposed system is presented. The authors present evidence of developing the federal exchange access control EHR framework in mobile clouds to investigate and validate the proposed plan. The performance and security of the proposed architecture are evaluated through two theorems.
The discussion section contains critical comments on scalability, availability, single point of failure, data integrity and security, and Performance Evaluation Comparison with Existing Methods.
The authors' findings are significant in terms of productivity and privacy protection.
It is seen that the diagrams of the figures and the inscriptions in them are cloudy and unreadable.
Author Response

(The authors gave the same response as above.)

Round 2
Reviewer 1 Report
Thanks to authors for the detailed response and additions I read through the comments and skimmed the revised PDF, The updates did improve the paper a lot. I would be happy to recommend this paper for publication |